# Partnership preferences, economic drivers, and health consequences of Gambian men's interactions with foreign tourists: A mixed methods study

Matthew Quaife[1]*, Mareme Diallo[2], Assan Jaye[2], Melisa Martinez-Alvarez[1,2]

**1** Faculty of Public Health and Policy, London School of Hygiene and Tropical Medicine, London, United Kingdom, **2** MRC Unit The Gambia at LSHTM, London School of Hygiene and Tropical Medicine, Serrekunda, The Gambia

* matthew.quaife@lshtm.ac.uk

**Data Availability Statement:** The cross-sectional survey data has been set to request access due to concerns about participant identifiability. The data

## Abstract

The Gambia has a thriving tourist industry, but in recent decades has developed a reputation as a destination for older, female tourists to seek sexual relationships with young Gambian men. During partnerships or in return for sex, Gambian men may receive financial support or in some cases the opportunity to travel to Europe with a partner. There has been little previous research among these men on sexual risk behaviours, physical and mental health, and health service utilisation. This study describes the economic drivers and health implications of interactions between Gambian men and foreign tourists near tourist resorts in The Gambia. We conducted simultaneous mixed method data collection among Gambian men who regularly interact with tourists: a cross-sectional quantitative survey and discrete choice experiment (DCE) with 242 respondents, three focus group discussions, and 17 in-depth interviews. The survey asked questions on demographic characteristics, sexual history and health-seeking, the DCE elicited trade-offs between partnership characteristics, and qualitative data explored individual and group experiences in depth. We found that sexual activity between Gambian men and tourists was prevalent with 50% of the sample reporting ever having sex with a tourist. Condom use at last sex was significantly higher with tourist (63%) than with Gambian partners (40%, p<0.01). Condom use, money, and opportunity to travel to Europe were most important to respondents in the DCE. Qualitative data validated and explained quantitative findings, notably pressures to engage in unprotected sex and potential travel to Europe. Although men's physical health needs were broadly met, mental health, substance use and sexual health needs were not. Young men working on the beaches of The Gambia face substantial health risks, including from STIs and mental health issues. The health system needs to understand barriers to existing health services, and how they can meet the needs of these vulnerable men.

set contains indirect identifiers (place of operation, age, gender, time period in location, time period working in industry) that might allow others, particularly those working in locations close to respondents, to determine the identify of a participant (or mistakenly attribute it to someone else). If a participant's identity was determined, their survey responses may have social, career or other implications for them or any colleagues mentioned. In addition, participants were told that information collected about them would be kept strictly confidential, and personal information would only be shared with certain individuals. The LSHTM ethics committee has advised that this does not allow for the data to be made publicly available. Data requests submitted through the LSHTM data repository (https://doi.org/10.17037/DATA.00003105) are sent to the project team and LSHTM Research Data Management Service for consideration. The LSHTM RDM Service acts as independent advisor for data access requests, directing them to the relevant ethics committee or other institutional body as appropriate. We encourage data access requests to be submitted through the LSHTM data repository, to ensure they're distributed to all named contacts. However, applicants can email researchdatamanagement@lshtm.ac.uk with the DOI for the dataset being requested if preferred.

**Funding:** Funding was provided through the Wellcome Trust through an Institutional Strategic Support Fund grant to LSHTM (Recipient: MQ). The funder had no role in the collection or analysis of data, or in the decision to publish.

**Competing interests:** I have read the journal's policy and the authors of this manuscript have the following competing interests: Since completing the study and after initial submission of this manuscript, MQ took a role at Evidera, a commercial research organisation providing for-profit research services to the pharmaceutical industry. MQ's work in this role is unrelated to the subject matter in this study. The authors have no further conflict of interests to declare.

## Introduction

Foreign tourism in The Gambia provides economic benefits, but also unique challenges. The Gambia is a popular package holiday destination for European tourists, and the tourist industry comprises around 20% of Gambian GDP [1]. Yet the impact of tourism is not unambiguously positive–in recent decades The Gambia has developed a reputation as a destination for older, mostly female tourists seeking sexual relationships with young Gambian men. This has been sensationalised in headlines of tabloid papers, referring to a "sex paradise" [2] with a "sleazy reputation" [3]. Although the close relationship between tourism and sex is not new [4, 5], there has been an imbalance in research to-date focusing on the sexual health needs of Western tourists and not their host destination partners [6]. Outside of The Gambia, the limited extant literature on populations hosting sex tourists has focused on narrowly defined commercial sex workers [7, 8].

During partnerships or in return for sex, Gambian men may receive financial support, and in some cases the opportunity to travel to Europe with a partner with the potential to settle permanently. Anecdotal descriptions that working in tourist-facing occupations on the Gambian coastline, including as hotel or bar staff, taxi drivers, or security guards, gives young men the opportunity to meet a European tourist and forge a relationship which may lead to opportunity for migration. A pseudo-occupation has developed for men seeking relationships with tourists, known colloquially (and to an extent pejoratively) as *bumsters*, or *chanters*.

To our knowledge, only one previous study [6] has been conducted among Gambian *bumsters*, which found:

"*Beach-boys, locally called bumsters, are a common feature of the country's tourism* [. . .] *the highly fantasized wealth [of Europe] forms the core of youth aspirations to travel abroad. Sexual activity with [tourists] is the ticket out of Africa's inherent scarcity*".

Such outward migration from The Gambia is common, and remittances from Gambians abroad make up around a fifth of GDP [9]. However, Gambians are an "important" source of irregular migrants travelling through the *backway*, a treacherous and illegal route to Europe through West Africa to Libya, and across the Mediterranean Sea. In 2016, 7% of irregular migrants arriving by sea in Italy, were Gambian [9], equating to roughly 0.6% of The Gambia's 2.2 million population.

It is notable that The Gambia sees older European women rather than men travelling for transactional sex, as only a few other countries see the typical gender roles of sex tourists reversed [10]. It is also notable that extremely little is known about the health risks facing tourists or the men they form relationships with. In a number of destinations renowned for men seeking transactional sex, health risks are well documented, for example in detailed epidemiological studies into sexually-transmitted infections (STIs) and condom use in Thailand, Cambodia, and China [7, 8]. Such data has led to the provision of specialist services by health systems and some non-governmental organisations, including widespread provision of condoms and reducing barriers to healthcare, for example through providing STI and HIV testing [7, 11]; this is not the case in The Gambia.

The absence of literature and preliminary work for this study suggested that the health needs of these men were unknown. Anecdotal evidence suggested that some men frequently engaged in risky transactional sex with tourists (male and female), and received no specialist healthcare to meet sexual, mental, or physical health needs. In addition, men were extremely economically precarious, relying on informal work entirely dependent on the tourist industry. In a setting where 25% of healthcare spending is out-of-pocket [12], a lack of ability to pay is

likely a substantial barrier to healthcare utilisation, resulting in avoidable morbidity and mortality.

Outside of the aforementioned anthropologic study [6], no research has been conducted in this population. The current study uses a mixed methods case study design to describe the economic drivers and health implications of interactions between Gambian men and foreign tourists near tourist resorts in The Gambia. We measure risk behaviours among Gambian men who form relationships with tourists and understand their causes and consequences. We then use a stated preference discrete choice experiment to explore men's preferences for partner and relationship characteristics, including trade-offs between risk and financial rewards. Finally, we measure healthcare access and care seeking, alongside key barriers to care.

This paper proceeds as follows: first we describe the study setting and design, the development of study tools, then their implementation. Then, we describe the methods used to analyse the quantitative and qualitative data. The results section is structured by theme: we use first quantitative and qualitative data to describe *bumstering* and the construction of our sample and the main activities of respondents. We then describe self-reported sexual behaviour and present results of a stated preference discrete choice experiment (DCE) assessing men's preferences for tourist partnerships, and then use qualitative data to add depth to these findings. Then, we present quantitative and qualitative data on other risk behaviours, including drug and alcohol use before, and finally describe health care needs and care seeking. The paper concludes with a discussion of the key findings, including key contributions of the study to the literature, and policy recommendations.

## Methods

### Study setting

Primary data collection was conducted in the main tourist areas of The Gambia's coastline, to the west of the capital Banjul. Three sites were chosen for recruitment along the stretch of coastline near Banjul home to a large number of hotels and tourist amenities such as restaurants, bars, and other attractions. The location of the sites is shown in Fig 1. The three sites were similar to each other in the type of mid-range hotel accommodation and amenities available, and preliminary work identified a number of potential participants at each site across registered and non-registered professions.

Preliminary work highlighted the extremely varied nature of occupations which led to potential relationships with tourists, alongside a perceived high prevalence of intention to and formation of sexual relationships with tourists among many men working in tourist industries. The target population for qualitative and quantitative data collection was therefore broad and included adult men working in tourist areas on the beach, in a profession which clearly meant they interacted closely with tourists. Although we anticipated that this approach was more sensitive than specific in identifying the target population, it was beneficial to a) capture data on a larger cadre of men with potentially unmet health needs and unique risk behaviours, b) have broad inclusion criteria to remove any stigma from participation among potential participants or others who observed the recruitment process, and c) gave interviewers operational inclusion criteria for an otherwise hard-to-define population. Interviewers and investigators worked closely throughout the study to ensure consistent definitions were being applied, including in debriefing conversations sessions where vignettes of potential participants were discussed.

Although partnership formation was perceived to be common and desirable across occupations, preliminary work identified a clear distinction in socioeconomic status and social standing between men working in formal and informal professions in the tourist industry–this also emerged during analysis of qualitative data. Those in the formal sector require registration and

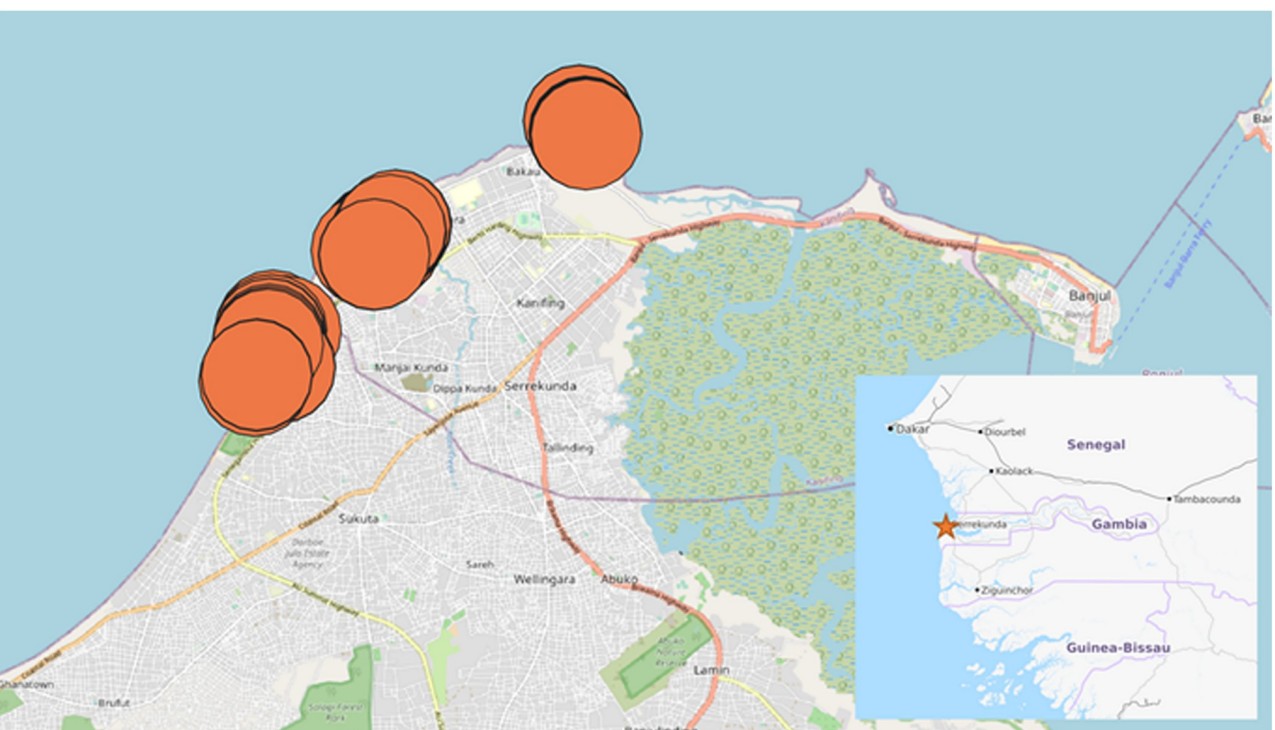

**Fig 1. Data collection locations.** Map data © OpenStreetMap (openstreetmap.org/copyright), used under a Creative Commons license, available at https://www.openstreetmap.org/#map=13/13.4183/-16.6817. Red circles display interview sites and are shown as randomly offset large circles to maintain anonymity.

membership of a professional association and included taxi drivers and tour guides. For the purposes of this study, we define the informal tourist sector as other occupations which require either no or minimal registration.

## Study design

This study has a mixed methods design with two qualitative and two quantitative components. The first quantitative component was a survey which sought to quantify risk behaviours and their correlates. The second quantitative component was a DCE which elicited the preferences of men for a range of tourist partner and relationship characteristics. The qualitative components comprised 17 semi-structured interviews and three focus group discussions (FGDs) and sought to add depth to quantitative measurement, and to capture dynamics not covered by quantitative tools. Qualitative and quantitative data were collected simultaneously and analysed sequentially, the steps of data collection and analysis are as follows. First, scoping discussions were held between the study team, ethics and scientific review committees, and two informal discussions held with potential participants; these discussions guided tool development and reaffirmed the value of the mixed methods approach. Qualitative guides and quantitative tools were developed and implemented, as described below. For analysis, qualitative transcripts were first read in a preliminary analysis–at this stage the dependent and independent variables for quantitative analyses were decided based on emergent findings, alongside the literature. The structure and content of quantitative analyses were informed by these initial qualitative analyses, specifically identifying independent variables and potential associations

with dependent variables. The thematic structure arising from preliminary qualitative analysis and the quantitative survey is used to present qualitative and quantitative findings.

## Tool development

The quantitative survey tool used standardised closed-ended questions in four sections: a) background and socioeconomic indicators, including income and food security, b) sexual history and risk behaviours, c) drugs and alcohol use, and d) health knowledge, care needs, and care seeking. Quantitative tools were piloted with five participants to check understanding and flow.

The DCE framework was based on a previous study eliciting preferences for sexual relationship and sex act characteristics [13]. Attributes and levels were selected based on preliminary discussions with men working with tourists, alongside researchers at MRC Unit The Gambia at the London School of Hygiene and Tropical Medicine (LSHTM). Five attributes were chosen due to their perceived importance in relationship formation and risk taking: condom use, money given by tourist to respondent, when money is given, the length of the relationship, and the age of the partner. An orthogonal, two alternative, unlabelled design was generated with an opt-out alternative of *neither*, and was piloted among four respondents in the target population of the quantitative study. The attributes and levels were deemed comprehensive by respondents, and pilot data were effects coded and analysed to give priors for an eight-task d-optimal efficient design for the final survey tool. Fig 2 shows an example DCE task from the survey tool, and Table 1 presents the full set of attributes and levels.

Qualitative in-depth interview and FGD guides were generated from preliminary discussions with men in the target population; two discussions took place to assess the feasibility the sampling and interview methods, and sensitivity of content. These preliminary conversations were not recorded and no data from these formally analysed. Guides were piloted in two interviews and finalised with minor changes to wording and question ordering; data from these interviews are included in analyses. FGD guides sought to elicit group attitudes and norms

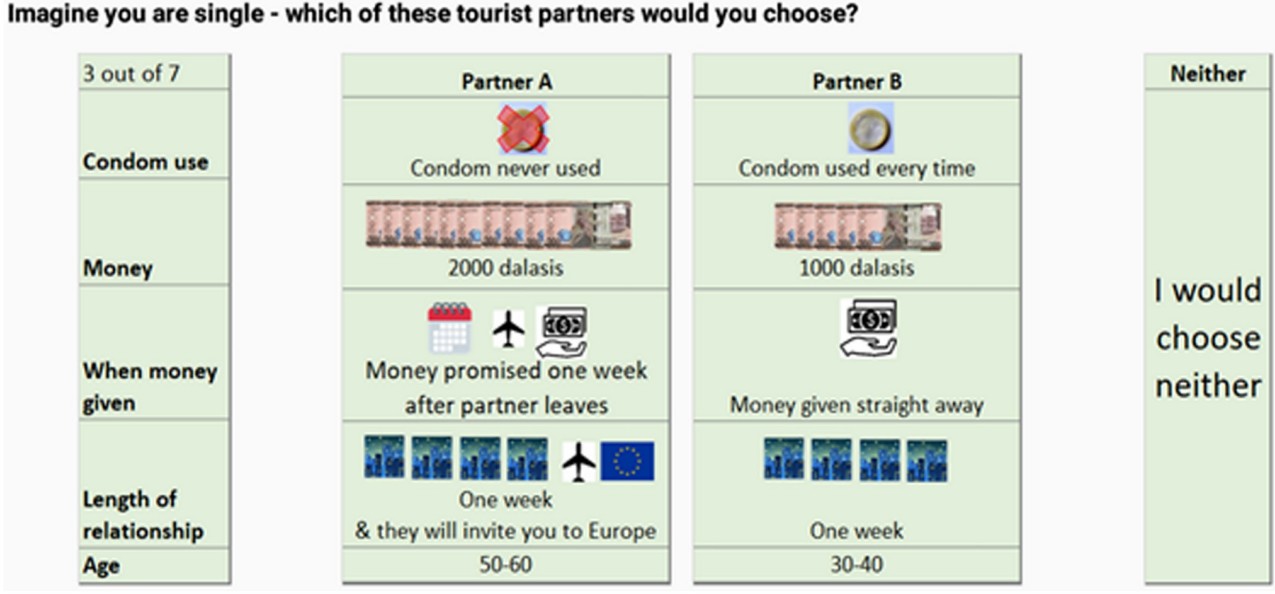

**Fig 2. Example DCE task.**

**Table 1. Discrete choice experiment attributes and levels.**

| Attribute | Levels | | | |
|---|---|---|---|---|
| **Condom use** | Condom used every time | Condom used half of the time | Condom never used | |
| **Money** | No money | 500 dalasi | 1000 dalasi | 2000 dalasi |
| **When money given** | Immediately after sex | Money given at the end of your partner's trip | Money promised one week after partner leaves | |
| **Length of relationship** | One night | 3–4 nights whilst partner is on holiday | Relationship lasts for their trip & partner will come back to see you in a few months | Relationship lasts for their trip & partner will invite you to Europe in a few months |
| **Partner age** | 30–40 years | 40–50 years | 50–60 years | |

around the context of relationships with tourists, whilst in-depth interviews sought to dig deeper into potentially sensitive topics around sexual relationships with tourist partners. The demographic information of qualitative participants was not collected, as in initial piloting this information was identified as a barrier to participation. The source guide for IDIs and FGDs is shown in S2 Text.

## Sampling and data collection

**Quantitative survey implementation.** Qualitative and quantitative data were collected simultaneously in April and May 2019. After one week of training, three Gambian fieldworkers from MRC Unit The Gambia at LSHTM conducted quantitative interviews using Open Data Kit (www.opendatakit.org) software on tablet computers. The DCE tool was part of the quantitative survey.

Qualitative and quantitative sampling was purposive. In each of the three study sites, interviewers spent one or two days familiarising potential respondents and business owners with the study and their intention to collect data in the area. Then, interviewers spent around one week to ten days in each site, seeking to interview all eligible men in the area who were willing to participate. Interviews were primarily conducted in English, which was widely spoken among participants who interact with tourists, though interviewers were able to clarify in Mandinka or Wolof if needed. Interviews generally took place outdoors in quiet areas of the beach where interviewers and respondents would not be overheard.

**Qualitative data collection.** The inclusion criteria for qualitative respondents were the same as quantitative work, though no participant participated in both qualitative and quantitative elements. For FGDs, a convenience sample of a small number of pre-existing social groups were sought to participate, and three FGDs were held with between three and six participants in each. Interview respondents were identified following the advice from the FGDs, either at the beach or in their work locations. Facilitators approached men who were sat in groups in tourist areas and asked if they were willing to participate in FGDs. The study and consent procedures were explained, and those not wishing to participate were asked to move out of earshot for the duration of the discussion. Participants in qualitative in-depth interviews were also recruited in the same manner as the quantitative survey. The interviewer explained the study and the consent process and found a quiet area where the interview could take place in privacy. FGDs and interviews were conducted by a Gambian interviewer primarily in English, though some Mandinka or Wolof phrases were used, were audio recorded and then translated and transcribed into English by a trained transcription team. As noted in the previous study among bumsters, FGDs were "well-suited to the male youth sub-culture in the Gambia, in which cliques ritually sit around a charcoal stove [. . .] smoking and chatting late into the night." On average, interviews lasted around 35 minutes, and the FGDs around one hour each.

## Data analysis

**Quantitative survey.** We first described the sample in terms of sociodemographic characteristics and the prevalence of risk behaviours. Based on preliminary analyses of qualitative data, and prior to quantitative analysis, we chose five indicators of risk behaviours across two domains as dependent variables in regression analyses, spanning sexual health and alcohol and drug use. In the domain of sexual health, we examine correlates of ever having sex with tourist, condom use at last sex, and reporting STI symptoms in the previous year. In the domain of alcohol and drug use, we examine correlates of reporting alcohol consumption and drug use in the previous three months. We fitted multivariable logistic regression models, using the same eight independent variables across models to analyse the determinants of risk behaviours consistently. These variables were also decided during preliminary qualitative analyses based on results from interview and focus group discussions and informed by similar correlational analysis in the empirical sex work literature from other contexts [13–15].

**Variable measurement.** Independent variables were not changed during quantitative analysis and were as follows:

*Participant age (years)*: Evidence from other settings suggests that younger men are more likely to engage in risk behaviours including risky sexual activity [16, 17] and alcohol and drug use [18–20].

*Registered occupation (yes/no)*: Formative qualitative data suggested that men working as taxi drivers and tour guides were qualitatively different to men with other tourist facing occupations. Registered occupations were hypothesised to give more income security and better access to tourists through reputable channels such as referrals from hotels and travel companies.

*Time working in industry (months)*: It was not clear *a priori* how time in the industry would affect different risk behaviours. Literature from other setting with transactional sex suggests that new entrants to sexual activities may be more vulnerable due to lack of knowledge or negotiating power [14, 21], though access to condoms may be easier than other locations [22]. Exposure to others using drugs and alcohol has been associated with engagement in these [19, 20].

*Months per year working in industry*: Formative qualitative work suggested a structural difference between men who had other sources of income such as farming and those who don't, though it was not clear how this would manifest in risk behaviours. Those with alternative sources of income may be less inclined to take risks in return for money, though may also have less experience of navigating risk in tourist areas.

*Difficulty in paying GMD 350 bill (yes/no)* and *medium or high household hunger (yes/no, versus low/no household hunger)*: These socioeconomic indicators have been found to be associated elsewhere with risk behaviours [17]. The ability to pay a medium sized bill has had internal and face validity in other settings as a proxy for income, which is difficult to measure accurately in surveys.

*Poor wellbeing (WHO scale, scored as "low" if under half possible score)*: A potential health indicator in its own right, poor wellbeing was chosen as a proximal determinant of risk behaviours based on preliminary work identifying poor mental health as a reason for engaging in drug use.

*HIV knowledge (number of items correct in nine item tool)*: A proxy for knowledge of sexual health risks, preliminary qualitative work indicated that myths existed around HIV transmission and treatment. The hunger, wellbeing, and HIV knowledge tools are provided in S1 Text.

**Choice model analyses.** We model DCE data using random utility models due to their consistency in explaining choice behaviour in health applications. We assume that that

individual $i(i = 1,...,N)$ makes choices such that they maximise utility over the three alternatives presented $(j = 1,2,3)$. Their axiomatic utility function $U_{ij}$ is decomposed into an explainable systematic component $V_{ij}$ and a random component $\varepsilon_{ij}$, and we specify an indirect utility function for the utility of respondent $i$ from choice $j$ in choice set $c$ as the linear combination of attributes and an error term:

$$V_{ijc} = X_{ijc}\beta + \varepsilon_{ijc} \tag{1}$$

With $V_{ijc}$ the utility derived from a choice, $X'_{ijc}\beta$ the component of utility that is captured by DCE attributes, and $\varepsilon_{ijc}$ a stochastic (random) component of utility. We specify the vector $X_{ijc}$ as the set of partnership attributes:

$$X_{ijc}\beta_j = \beta_0 + \beta_1 condom\_use_j + \beta_2 money_j + \beta_3 time\_of\_payment_j \\ + \beta_4 relationship\_length_j + \beta_5 age_j + \beta_6 opt\_out_j \tag{2}$$

Where $condom\_use_j$, $money_j$, $time\_of\_payment_j$, $relationship\_length_j$, $age_j$, and $opt\_out_j$ the design attributes of the DCE, and $\beta_0$ a constant. We first estimate equation [1] using a multinomial (or conditional) logit model (MNL) which estimates the probability of individual $i$ choosing alternative $j$ among the set of options $c$ as a probabilistic function of design attributes:

$$P_{ijc} = \frac{\exp\left(X_{ijc}\beta\right)}{\sum_j \exp(X_{ijc}\beta)} \tag{3}$$

The MNL requires two potentially restrictive assumptions: the IIA assumption of independence of irrelevant alternatives (concordant with the IID distribution of the disturbance), and homogenous preferences across individuals. We therefore also run mixed multinomial logit (MMNL) models to which goes some way to avoiding these limitations through the inclusion of random taste heterogeneity; we run MMNL models with 1000 Halton draws using normal distributions for all parameters. Results are shown in the form of relative utility weights in figure forms, and willing-ness-to accept analyses conducted with price as the denominator.

**Qualitative analyses.** A schematic of the qualitative and quantitative analysis is shown in Fig 3. For qualitative analysis we developed a coding framework based on the initial open coding described above and the emergent patterns in the quantitative data. First, we identified three codes which would necessarily arise from the structure of the quantitative tools. We then conducted an inductive coding process by analysing transcripts and listing all codes before imposing this superstructure as a basis for the coding tree, with three of the authors reading one transcript to extract codes and validate the process, then dividing up remaining transcripts for coding. Qualitative results are presented alongside quantitative results in the same thematic domain. NVIVO 12 software was used for analysis, transcription was verbatim and random quality checks conducted by transcription supervisors.

## Ethical considerations

The study protocol and tools were approved by the London School of Hygiene and Tropical Medicine Observational/Interventions Research Ethics Committee (16137), The Gambia Government/MRCG Joint Ethics committee (1638 V1.1), alongside the Scientific Coordinating Committee of the MRC Unit The Gambia at LSHTM.

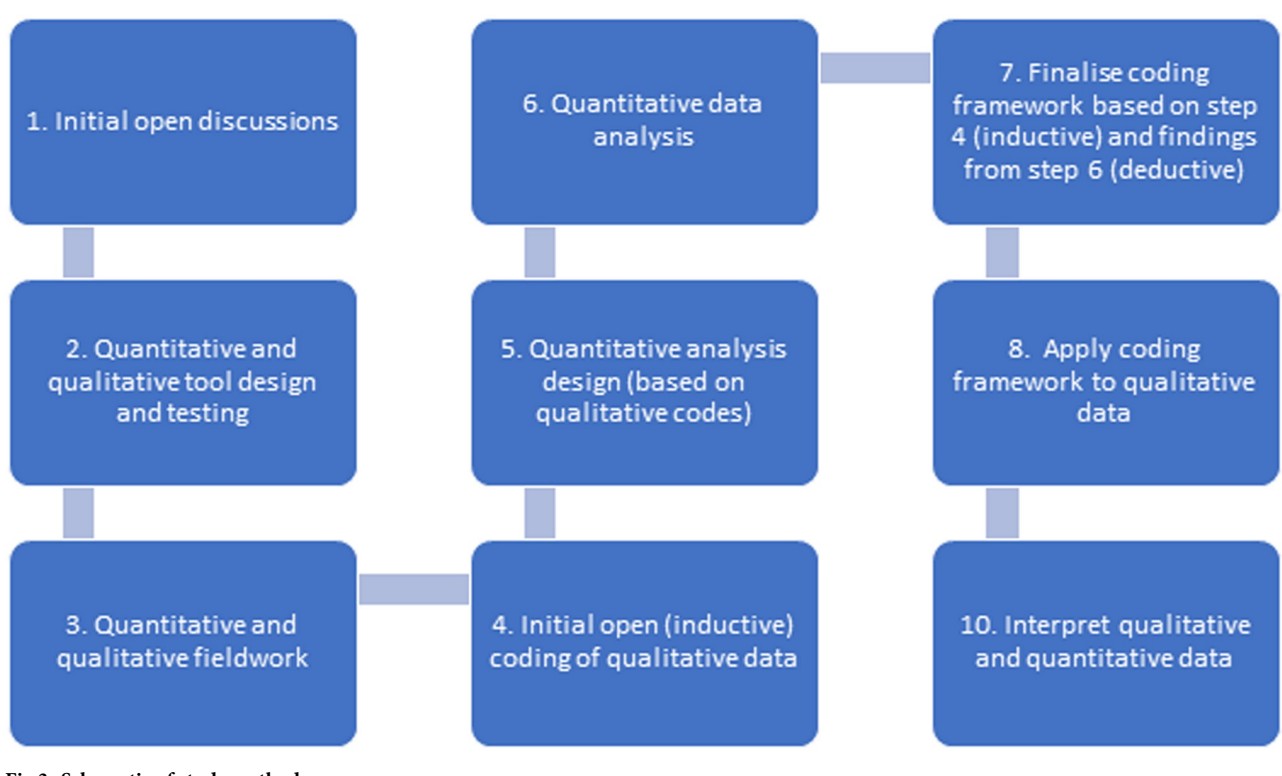

**Fig 3. Schematic of study methods.**

## Results

Results are presented in four sections, each with qualitative and quantitative elements: a) descriptions and characteristics of bumsters, b) risk behaviours and their determinants, c) preferences for tourist sexual partners and relationships, and d) healthcare needs and care seeking.

### Descriptions of bumsters and activities

Table 2 presents the descriptive statistics of the quantitative sample. The mean age of 39 was older than the median age of the Gambian population (20 years). A high proportion (85%) reported difficulty in paying a GMD 350 (~GBP £5.37) bill and reported a monthly income of GMD 3 468, (~GBP £53). Around 15% reported experiencing medium or high household hunger. 40% of respondents worked in the registered, formal sector as a tour guide or taxi driver. Of the remainder, 11% were juice pressers, 8% self-identified as bumsters, and 29% reported a different occupation. On average, men had spent around nine years working with tourists, and many worked for the entirety of the tourist season, a mean of 7.4 months per year were spent working with tourists.

When asked to define a *bumster*, respondents' descriptions varied, though it was apparent that a key activity of a *bumster* is to approach and communicate with tourists, primarily to direct their attention to a business or forge a friendly relationship from which money or goods would later arise.

There was a recognition such communication was a basic part of all jobs and that bumsters are everywhere. A number of participants said "*everyone is a bumster*"–from people selling things on a beach to registered guides showing tourists attractions. A common refrain was

**Table 2. Descriptive statistics of quantitative sample.**

| | Mean | SD | Median | N | % | Obs. |
|---|---|---|---|---|---|---|
| Age | 39.2 | 9.58 | 38 | | | 242 |
| Secondary education as highest education level | 0.7 | 0.46 | | 169 | 70% | 242 |
| Would find it more than a little difficult to pay 350GMD bill | 0.85 | 0.35 | | 206 | 85% | 242 |
| Occupation | | | | | | |
| *Tour guide (registered)* | | | | 48 | 20% | 240 |
| *Tourist taxi driver (registered)* | | | | 48 | 20% | 240 |
| *Juice presser* | | | | 27 | 11% | 240 |
| *Bumster / chanter / chancer* | | | | 20 | 8% | 240 |
| *Restaurant / bar worker* | | | | 10 | 4% | 240 |
| *Horse rider* | | | | 5 | 2% | 240 |
| *Fruit seller* | | | | 3 | 1% | 240 |
| *No occupation* | | | | 9 | 4% | 240 |
| *Other* | | | | 70 | 29% | 240 |
| Monthly income (GMD) | 3468 | 3960 | 3000 | | | 242 |
| Time working with in industry (months) | 110 | 97 | 90 | | | 242 |
| Months per year working with tourists | 7.4 | 2.67 | 6 | | | 242 |
| Moderate or high household hunger | | | | 35 | 15% | 240 |
| **Sexual history** | | | | | | |
| Currently cohabiting | | | | 123 | 51% | 242 |
| Age at first sex | 30.4 | 45 | 22 | | | 242 |
| Number of lifetime sexual partners | 5.4 | 6.37 | 4 | | | 239 |
| Ever had sex with tourist | | | | 122 | 50% | 242 |
| Condom use (last sex) | | | | 95 | 39% | 242 |
| Condom use (always) | | | | 74 | 31% | 242 |
| Source of last condom | | | | | | |
| *Pharmacy* | | | | 49 | 52% | 95 |
| *Friend/relative* | | | | 25 | 26% | 95 |
| *My partner gave it to me* | | | | 11 | 12% | 95 |
| *Other shop* | | | | 2 | 2% | 95 |
| *Health clinic (public)* | | | | 3 | 3% | 95 |
| *Health clinic (private)* | | | | 2 | 2% | 95 |
| *Other (Specify)* | | | | 2 | 2% | 95 |
| Age difference of last sexual partner | 7.8 | 9.8 | 8 | | | 241 |
| Origin of last sexual partner | | | | | | |
| *Live here—of Gambian origin* | | | | 163 | 67% | 242 |
| *Visitor—came for a short period of time* | | | | 45 | 19% | 242 |
| *Live here—of other origin* | | | | 27 | 11% | 242 |
| *Visitor—but stay in the Gambia for long periods* | | | | 2 | 1% | 242 |
| *Don't know/other* | | | | 5 | 2% | 242 |
| Number of tourists had sex with | 2.2 | 3.06 | 1 | | | 122 |
| Condom use (last sex with tourist) | | | | 75 | 61% | 122 |
| Condom use (always with tourists) | | | | 64 | 52% | 122 |
| Number of acts with last tourist | 4.9 | 6 | 3 | | | 122 |
| Received gifts or money in return for sex | | | | 43 | 35% | 122 |
| Relationship maintained after partner left | | | | 73 | 60% | 122 |
| **Health and health seeking** | | | | | | |
| STI or symptoms in last 12 months | | | | 22 | 9% | 242 |

*(Continued)*

**Table 2.** (Continued)

| | Mean | SD | Median | N | % | Obs. |
|---|---|---|---|---|---|---|
| Ever taken HIV test | | | | 91 | 38% | 242 |
| >1 alcoholic drink in last 30 days | | | | 42 | 17% | 242 |
| Drank heavily at last sex | | | | 12 | 5% | 242 |
| Drug use in last 3 months | | | | 115 | 48% | 242 |
| Mental wellbeing (WHO-5)—total | 16.2 | 4.2 | 16 | | | 242 |
| Mental wellbeing—<50% aggregate score | | | | 56 | 23% | 242 |
| Highly risk-loving (10/10) | | | | 139 | 57% | 242 |
| HIV knowledge—v high (9/9) | | | | 60 | 25% | 242 |
| HIV knowledge—low (<6/9) | | | | 113 | 47% | 242 |
| Last care location | | | | | | |
| *Government hospital* | | | | 82 | 34% | 242 |
| *Government health centre* | | | | 53 | 22% | 242 |
| *Pharmacy* | | | | 36 | 15% | 242 |
| *MRC Fajara* | | | | 30 | 12% | 242 |
| Private hospital | | | | 17 | 7% | 242 |

"*even the president is a bumster*", as he communicates with other leaders to bring economic prosperity to The Gambia. All of these activities were viewed in a non-judgmental way and not morally inferior to other means of earning money and providing for families. Two participants identified interviewers themselves as bumsters:

" *Even you are a bumster. You have approached me and talking to me. . .all of that is bumsing. You woke up in the morning with the purpose that you must do this today. . . It's all bumsing*"

[interview participant]

Poverty and a lack of other economic alternatives was the unambiguous reason for *bumstering*. There was a clear sense of opportunity at the beach, that *bumsters* come to make something of themselves from an otherwise disadvantaged position.

"*Everybody want to survive . . . everybody wants to live good. . .you know. . .everybody wants to make his parents happy [. . .] We have to work for it but no employment. . . you have to come to the beach. . .you hustle. . .. you meet good friends; maybe you can be lucky they help you*"

[Male interview participant.]

Respondents also mentioned that if they had jobs, youths would not go to the beach. They wished the government would create jobs and invest in skills and education of young boys. The alternatives to the beach are seen as selling drugs or begging in the street, which were deemed less desirable than *bumsing*.

"*Before going about in town stealing or selling marijuana, it will be better to come here. Because if you are caught in selling marijuana your family will go through a stress. You will get stressed likewise your family. If you should be caught in a robbery maybe it will be a stress for only you. it could be your family may also get involved in the stress*"

[interview participant]

In the short term there is the possibility of money whilst tourists are present, and a good tourist friend will, in the medium-term after they return home, send phones or other valuables which can be sold or used productively. A large proportion of respondents described how tourists had sent money, mobile phones, cars, or bought property for Gambians. There was also some perceived status to having a white partner:

"*But seeing it as being with a white lady, whether she gives you anything or not, just for people to see that you have a white lady. . .that is why some others do it. Some will say I'm doing it so that I can be capable to do somethings in the future. Driving big cars or I will be travelling. Or I will get a compound, or my life will be transformed. . .that is why some do it*"

[interview participant]

A number of interviewees described such external help as a matter of survival, and good friends (defined as those who send such items regularly) seen as facilitating a move out of *bumstering* to run businesses based on these assets. Sometimes friendship is in the form of dating, and the rationale for dating older women was described in purely economic terms. The possibility of travel to Europe was an important factor.

Respondents noted that, just as there are good and bad people everywhere, there were also good and bad *bumsters*. Good *bumsters* were described by their intention to build long term relationships with tourists to develop mutually beneficial friendships—providing friendship and hospitality, in the hope that a longer-term relationship may lead to cash or other valuable assets. Relationships between good *bumsters* and tourists were framed by respondents primarily as a means of ensuring that tourists enjoy their trip, and in return they give men a payment of what the tourist deems fair. Gambian partners and families of bumsters also benefit from the money that relationships with tourists bring.

"*If you build an acquaintance with a white man and he/she knows about your life history. Because at times if your friendship went on for long, you can take him/her to your home to meet your mom and dad. When going for that family visiting, they will buy a bag of rice, a gallon of oil, a bag of onion, a bag of potatoes. . .they will buy chocolates for the kids. All these to visit your home and make the family happy. Family visit. . .that is what we call family visiting*"

[interview participant]

Bad bumsters were described as cheating tourists, and only making relationships with tourists in the short term, looking to make or steal money:

"*You went shopping with a white man in a market and you tell him the cost of something is D1200. Hence, he has no idea, he will give you D1200 when in reality the cost is D200. That one is a bad bumster. In turn the white man gets to learn that he had been misled and then the relationship between him and the bumster will be destroyed.*"

[interview participant]

One respondent described falsely telling a tourist partner that he was not married. The tourist moved into his home for a while, living with his Gambian wife who was falsely described as his sister.

This was reflected in discussions on how the government should treat bumsters, where the majority of respondents felt that, given they made a positive contribution to the Gambian

economy by helping tourists, the government should regulate them, provide accreditation, and they should form "groups with a leader". This was in contrast to their reality, where hotels and police would kick bumsters off the beach, particularly if they are making noise. Having a registration card was also suggested to protect bumsters from hotels and the police.

Good tourists were described in innocent terms, as people who help Gambians without using them, toying with them, or trying to have sex with them. A good tourist is responsive to demands, asking what people want then providing it. Bad tourists were primarily identified in two ways, the first linked with alcohol or drug use, including forcing Gambians to use these. Secondly, and more pragmatically, a bad tourist was also described as someone who can't transform your life because they are too poor or not generous enough, or who promises things they do not deliver:

> "Some as I said can transform your life. . .but you can be with one for twelve or thirteen years and they wouldn't do anything for you. Sometimes when coming. . .see you later. . .see you later. . .maybe when going back they will give you the remnants of their body creams or clothes [. . .] you have been with the person for a long time and they has still not done anything for you. They is not good."

[interview participant]

The mood of tourists was described as being changeable, which was problematic for *bumsters* who are totally dependent on them. Payments after tourists leave were particularly important in the off-season. Alongside regular financial or asset transfers, men had high hopes that relationships with tourists would lead to the opportunity to travel to Europe–described in the previous study with this population as "a land of milk and honey" [6]:

> "They expect that they get invitation to come [to Europe]. They expect that they make some money every month. They expect marriage all those to get better life."

[interview participant]

> "Some boys were there that was working along the beach. . .you know. They have some friends; they took them out of this country. They go to Europe. When they come back, they see them change you know. That what they exciting. . .give them . . .to dream of going abroad. you know what I mean. And the way for them is to be close to the tourists. You know. To be close to them"

[interview participant]

> "May you have a good. . .good white woman or white husband is what everybody is praying to people you know. [. . .] <because> they are the only ones who can help you"

[interview participant]

## Sexual behaviour

Sexual activity with tourists was prevalent; 50% of respondents reported ever having a sexual relationship with a tourist, with a mean of 2.2 sexual relationships with tourists among those reporting any. These relationships were relatively short–the average number of sex acts was 4.9 –and self-reported condom use was significantly higher at last sex with a tourist partner compared to Gambian partners (61%, diff: 22 percentage points, p<0.01), and "always" with tourist

partners (52%, diff: 22 percentage points, p<0.01). Interviews and FGDs confirmed that Gambian men engaged in sexual relationships with tourists. 33% reported receiving gifts or money directly in return for sex. One in five last sexual partners were visitors to The Gambia, and 60% of relationships were maintained in some way after the partner left.

Compared to other surveys of groups perceived to be at high risk of STIs, sexual activity as measured by number of lifetime partners (5.4), age at first sex [30], and age differential of last sexual partner (7.8 years) indicated lower risk than other key population surveys in sub Saharan Africa e.g., [23, 24]. 9% of respondents reported STI symptoms in the previous year, and 38% ever having taken a HIV test.

## Preferences for tourist sexual partners and relationships

In this section, we first present and interpret multinomial logit results of DCE choice data in Fig 4, then present qualitative data from these themes. Tabular results for different choice models are shown in, Table B in S1 Table. DCE results demonstrate face validity as coefficients are of the direction and relative magnitude as expected. DCE levels are effects coded, meaning that utility weights greater than zero imply a positive preference, and below zero a negative preference.

The most influential attribute on respondent choices was condom use, with "always used" strongly and significantly preferred (β = 0.679, p<0.01) with "never used" significantly disliked to a similar extent (β = -0.710, p<0.01). Money was the second most influential attribute, and although point estimates indicate non-linear preferences, models with a continuous money

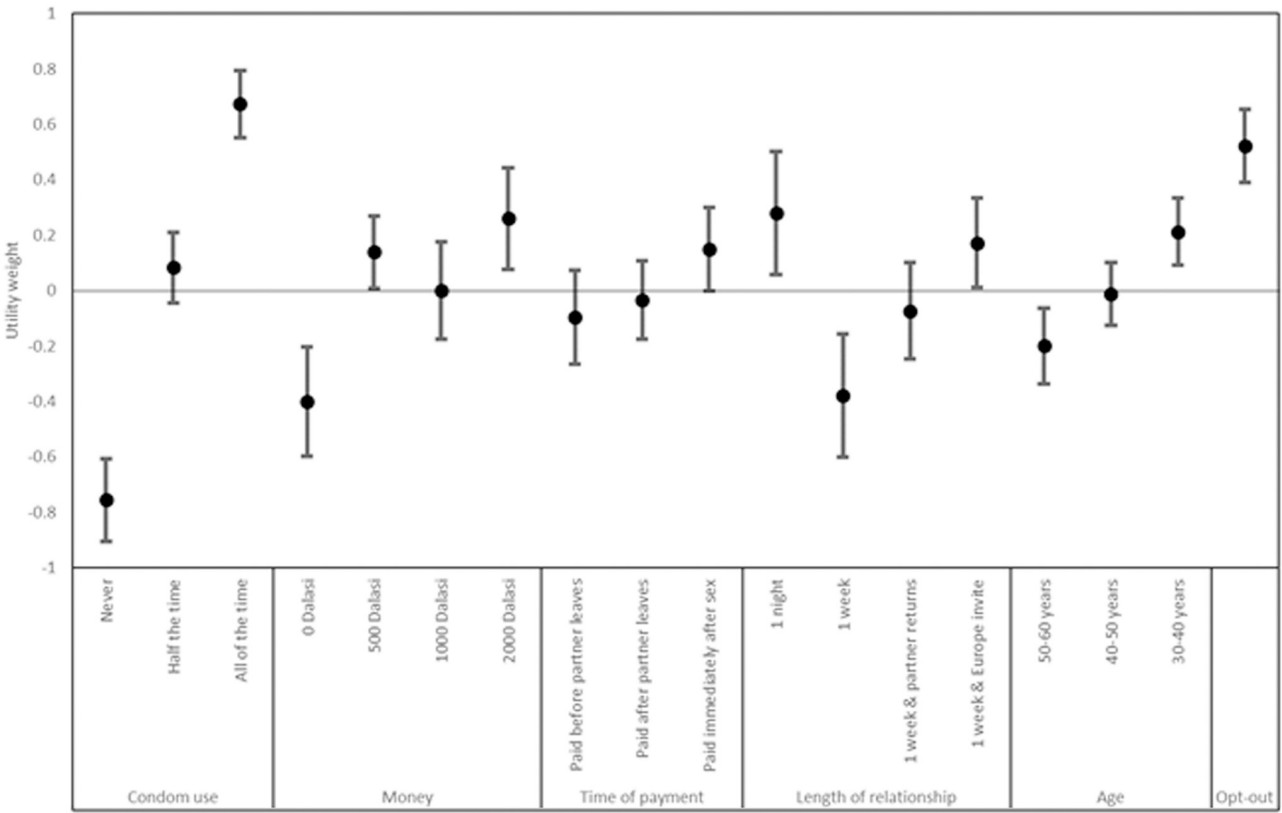

**Fig 4. DCE results—Effects coded multinomial logit model.**

attribute in (models 2 and 3 in Table B in S1 Table) show a significant and positive preference for money transfers (p<0.01). Respondents favoured money transfer taking place immediately after sex (β = 0.132, p = 0.05). A one-night relationship was favoured the most, followed by the potential for an invitation by a partner to Europe (β = 0.255, p = 0.016), and 3–4 day relationship during a partner's holiday was disliked (β = -0.379, p = 0.015). Finally, younger partners (30–40 years, β = 0.212, p<0.01) were preferred compared to older partners (50–60 years, β = -0.200, p<0.01). A positive and significant opt-out parameter shows that, all else equal, respondents would prefer not to engage in relationships with tourists (β = 0.522, P<0.01).

Willingness-to-accept estimates for statistically significant parameters in the model with a continuous price attribute (model 2 in Table B in S1 Table) indicated that respondents required GMD 3532 (GBP £50.46) to compensate for the utility loss of condoms never being used, and GMD 2050 (GBP £29.28) to compensate for utility loss of the relationship with the tourist in general.

Just as the DCE shows a willingness to trade condom use against other tourist and relationship characteristics, qualitative data also showed that condom use was not ubiquitous. For example, respondents from one FGD had the following exchange when asked about condom use:

> "Interviewer: *So, do you think people here use protection or when they sleep with the tourists*?
>
> Participant 4: *yeah, yeah if you love yourself. . .*
>
> Participant 2: *not many*
>
> Participant 3: *it depends*
>
> Participant 3: *if you are a wise guy you will use it*
>
> Participant 2: *like some people like it. . .some don't like it*"
>
> [FGD 1, four male participants]

Overall, respondents had good knowledge of why condoms were used and where they could be obtained. Several mentioned not being able to tell someone's disease status by looking at them, and hence the need to wear a condom to prevent STIs.

> "*Because everybody is aware that there are diseases about. . .you cannot just see a person like that and make a connection with her when you don't know the kind of person she is. You don't know what she is carrying [. . .].When connecting with [tourists] and you don't know each other, we use condom*"
>
> [interview participant]

The Gambia is a majority Islamic country with a Christian minority. Qualitative data show that respondents could not always follow all of Islam's teachings. For example, religion was also mentioned as prohibiting the use of condoms, however, not having children outside of marriage was also not permitted, which was given as a reason for wearing condoms.

> "*You should be very mindful, even not about white girls, women in general. If you are with one that you don't know, protect her and know that the fornication you are doing is detested by God. If you are mocking God and if you begot a child from that, that will be an abomination. So, can you protect yourself if you can lay hands on a condom*"
>
> [interview participant]

Quantitative data showed that around half of condoms were obtained from pharmacies, whilst 12% were reported to have been provided by partners.

Women's preference was deemed an important factor in the decision to wear a condom:

"*You see those women. When going they go with everything they need. In their purses. They carry a packet of condom. If you claim to not have a condom, they would be ready [. . .] wherever you try them they will be with a condom.*"

[interview participant]

Although most respondents mentioned women preferring condoms, one respondent mentioned not wearing a condom because women asked him not to. Several other reasons were given for why condoms were sometimes not used. In one FGD, a respondent also mentioned that sex is "*less pleasurable*" with a condom. In a few interviews, respondents suggested variable condom use with tourist partners was in part due to the quick advance to sexual contact in tourist relationships:

"*. . .some don't use to bother about [safe sex] [. . .] They will be ready to do it at the very spot. They can have you bend down at the beach itself*"

[interview participant]

Sometimes *bumsters* have sex with male tourists. This was mentioned in some interviews, where respondents reported negative feelings towards this practice. We note that the translation of gendered words to English was often imprecise, for example, "he/him" was frequently used to describe a woman, so this inference was from specific descriptions of homosexual activity. Participants attributed homosexual behaviours as being introduced in The Gambia by white tourists, and once the tourists leave these behaviours persisted:

"*That is men doing anal sex with each other. [. . .] That is why tourism is good for a place and at the same time bad for it. Because it destroys the place. Because the habit they come with, some of it is left here when they leave.*"

[FGD 1, four male participants]

Knowledge of STI screening and services was generally high, with STI screening at the start of a new relationship with Gambian partners or tourists mentioned as a key risk mitigation strategy:

"*Me my wife <who is from UK>, before we have anything, we went to hospital. We check everything. They test my blood. They test everything. And that night we came we make things happen.*"

[interview participant]

"*Yes if there is no marriage between you, then you should not go intimate with her without going to the doctor.*"

[interview participant]

Tourists who sleep with lots of different men perceived to pose a higher risk, but also to bring little benefit to the bumsters themselves.

"*This lady is risk what I see this lady. She can be a poor lady ever. She can be a poor lady; she can be a lady with sickness. She can bring good luck for the beginning. Bad luck in the ending*"

[interview participant]

## Drug and alcohol use

Almost half of the quantitative sample (48%) reported drug use in the previous three months, in all cases specifying marijuana as the sole drug used. Qualitative data suggested harder drugs were often used. 17% report drinking alcohol in the previous 30 days, and 5% drinking heavily at last sex.

The relationship between being a *bumster* and the consumption of alcohol and drugs is reciprocal. Just as *bumster* activity can lead to the use and addiction to alcohol and drugs, so can substance use increase the risk of becoming a *bumster*. The risks from tourists facilitating introductions to hard drugs and alcohol were identified, and participants specifically noted that the power dynamic from monetary transfers influenced the behaviour of men:

"Participant 2: *You know. . . and they drink it <drugs> with alcohol. . . some will be drug addicts after because they link with bad white guy or white woman or a white girlfriend. . .who will introduce them into these hard drugs. . .you know*

Participant 3: *and they cannot control their feelings because they are poor. . .they never receive D100,000 [before]. . .you give them D100,000, they want to follow you in your doing things*"

[FGD 1, four male participants]

"*No, I won't refer them as good tourist. . .such are not good friends (. . .) And also, there are some (. . .) you should not let yourself get into their systems. . .Because maybe they deal with powdered drugs (probably cocaine) and the likes(. . .) If you get in to their system, they can initiate you into things that will destroy you, and you will not be able to access it (. . ..) Do not let that happen. . ...Even there are some white women. . ..do not get into their systems*"

[interview participant]

In another FGD, the risks of alcohol use were starkly described:

"*Participant 2: Like many bumsters lose their lives here*

*Participant 1: Many bumsters there you see before last year, some bumsters dead on the beach*

*Participant 2: Because they spend more than fifteen years, twenty years on the beach you know just to try and get something*

*Participant 1: He is drink drink drink, he is dead*"

[FGD 1, four male participants]

## Correlates of risk behaviours

Multivariate analyses of correlates with risk behaviours are shown in Table 3. There was reasonably strong evidence that being in the registered occupations of tour guiding and taxi driving was potentially protective, associated with lower odds of STI symptoms in the previous year (OR: 0.24, 95% CI: 0.1–0.8), alcohol use (OR: 0.4, 95% CI:0.2–0.8) and recent drug use

**Table 3. Correlates of risk behaviours: Multivariable logistic regression.**

| | [1] | [2] | [3] | [4] | [5] |
|---|---|---|---|---|---|
| | Ever had sex with tourist | Used condom at last sex | STI symptoms in previous year | Drinks alcohol | Used drugs in previous 3 months+ |
| Age | 0.978 | 0.980 | 0.949 | 1.022 | 0.920*** |
| | (0.0171) | (0.0181) | (0.0352) | (0.0185) | (0.0288) |
| Registered occupation | 0.733 | 0.751 | 0.248** | 0.422*** | 0.431* |
| | (0.204) | (0.216) | (0.148) | (0.124) | (0.216) |
| Difficulty in paying GMD 350 bill | 0.960 | 0.694 | 1.890 | 0.891 | 2.463 |
| | (0.365) | (0.270) | (1.574) | (0.353) | (1.494) |
| Time working in industry (months) | 1.005*** | 0.999 | 0.999 | 0.997* | 0.990*** |
| | (0.00188) | (0.00188) | (0.00433) | (0.00197) | (0.00363) |
| Months per year working in industry | 0.906* | 0.895** | 0.997 | 1.131** | 0.831** |
| | (0.0469) | (0.0497) | (0.0925) | (0.0595) | (0.0742) |
| Poor wellbeing (<half) | 0.867 | 0.859 | 0.337 | 0.935 | 3.455* |
| | (0.279) | (0.293) | (0.270) | (0.323) | (2.281) |
| Medium or high household hunger | 0.957 | 1.089 | 4.488*** | 1.316 | Omitted++ |
| | (0.364) | (0.424) | (2.356) | (0.518) | |
| High HIV knowledge | 0.972 | 1.147** | 1.149 | 0.959 | 1.112 |
| | (0.0562) | (0.0708) | (0.127) | (0.0588) | (0.113) |
| Constant | 0.978 | 0.980 | 0.949 | 1.022 | 0.920*** |
| | (0.0171) | (0.0181) | (0.0352) | (0.0185) | (0.0288) |
| Observations | 240 | 240 | 240 | 240 | 130 |
| R-squared | 0.042 | 0.066 | 0.109 | 0.092 | 0.333 |

Odds rations from multivariable logistic regression are reported. Standard errors in parentheses.

*** p<0.01,

** p<0.05,

* p<0.1.

+ Only those reporting ever using drugs (n = 130) included.

++ All those who reported drug use in previous 3 months also reported medium or high hunger.

(OR: 0.4, 95% CI: 0.2–1.1). All those who reported drug use in previous 3 months also reported medium or high hunger. These results align with qualitative findings that men in registered occupations may be less likely to engage in risk sexual relationships. High HIV knowledge was positively associated with condom use at last sex (OR: 1.147, 95% CI: 1–1.3), as expected *a priori*.

However, other factors have inconsistent correlations with risk factors. A greater dependence on tourism, measured by the number of months per year working in the industry, was marginally associated with lower condom use at last sex (OR: 0.9, 95% CI: 0.8–1) and alcohol use (OR: 1.1, 95% CI: 1–1.3), but lower odds of ever having sex with a tourist (OR: 0.9, 95% CI: 0.8–1) and drug use (OR: 0.83, 95% CI: 0.7, 1). Experience of household hunger was associated with STI symptoms in the previous year (OR: 4.5, 95% CI: 1.6–12.6).

## Healthcare needs and care seeking

Fig 5 demonstrates that almost all respondents–regardless of whether they have sexual relationships with tourists–reported one or more health needs in the previous 12 months. Whilst physical health needs were broadly met, mental health, substance use and sexual health needs

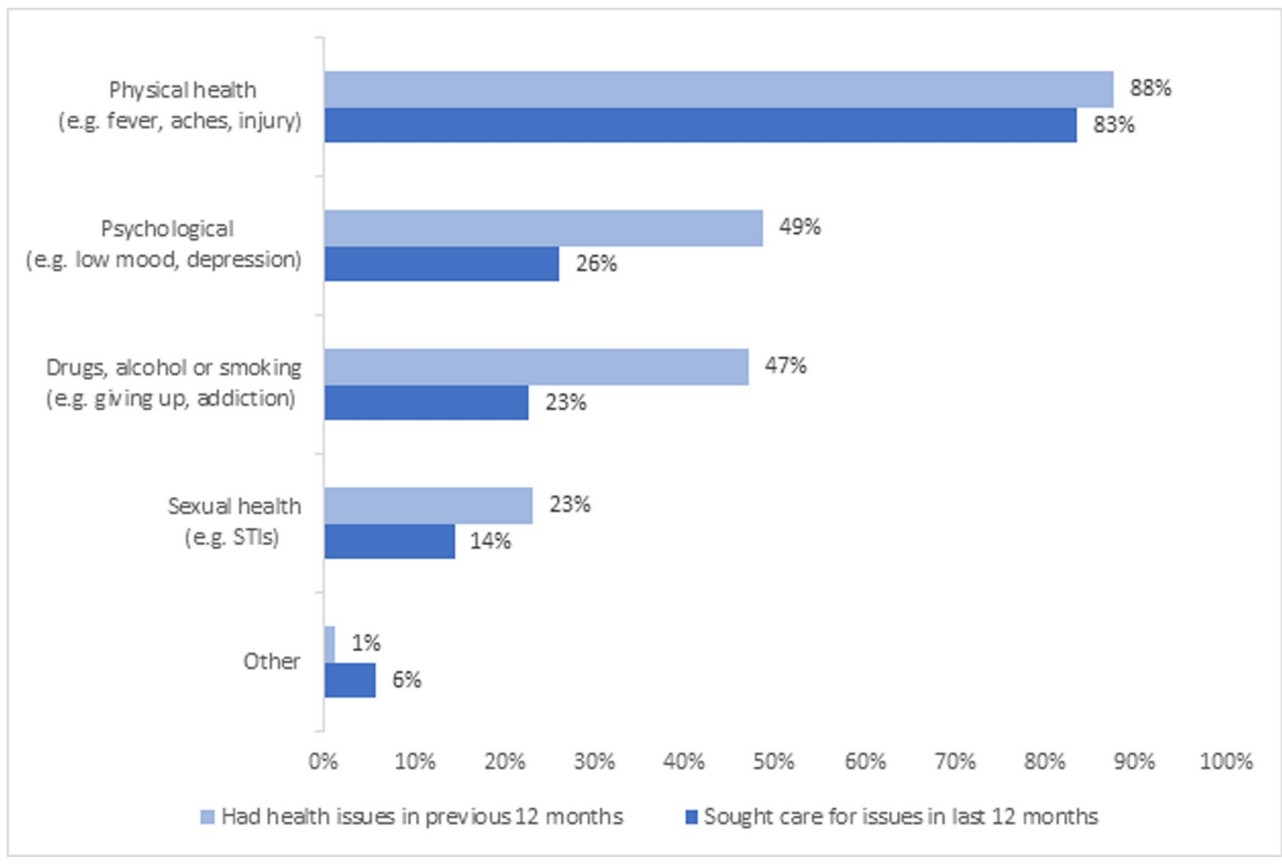

**Fig 5. Healthcare needs and care seeking in previous 12 months.**

were not. Nearly nine out of ten (88%) respondents reported physical health needs in the previous 12 months, with almost all (94%) of those with needs reported having sought care. For psychological health needs or assistance with drugs, alcohol, or smoking, although 49% and 47% of respondents had experienced respective needs in the previous 12 months, only around half sought care to meet needs. Just under one quarter (23%) of respondents reported sexual health needs in the same timeframe, and 61% sought care. It was notable that the 23% reporting sexual health needs is over double the prevalence of self-reported STIs in the previous year (9%), perhaps reflecting a broader sexual health need than only infection.

Qualitative data also demonstrated a general recognition that preventive and curative care was available, and that bumsters were at increased risk of infections because they interact with a lot of people. There was an acknowledgment that everyone gets sick–"*even the president*". Although some described going to hospital as easy, others identified challenges to healthcare access in rural areas where transport was a key barrier. Alongside not having money to pay out-of-pocket for care, stigma from STIs or other embarrassing conditions was identified as a barrier.

Qualitative data also illustrate barriers to seeking sexual health care. Obtaining HIV testing was said to bring shame to the patient: "*some will prefer to die instead of talking about it*". Making use of social ties could be a good way to manage stigma or shame. One respondent explained that if a person is ashamed to use a health service for fear of being recognised, they can go through an acquaintance who works there:

"*Maybe you might be shy to tell them that is what you there to investigate on. But if you know someone working at [the health facility] you can relate your difficulty to them. Okay, maybe that person could know someone in the section of those in charge of that and refer you to that person. you see. But maybe if you go there and be interviewed might be difficult for you.*"

[FGD 1, four male participants]

There was a perception of poor quality care in public facilities, including with long waiting times and unavailability of drugs: "*only paracetamol is available*". Private healthcare was described as a quicker and better quality option, though expensive and therefore not accessible for everyone. The staff of public facilities were perceived to be inattentive to patients' needs:

"*But I used to see people standing. . .patients will be sitting without being considered by the doctors. A doctor should have his/her breakfast early in the morning before the patients come around. You should concentrate on the work you are doing to keep going so that it moves on. So that when the patients come they will see that there is truly a doctor here. But if you go there. . .the cases that I have witnessed, have being very moving to me*"

[interview participant]

Self-medication without seeking care first was thought to be prevalent. Some participants mentioned the use of herbal or traditional medicine as an alternative to western medicine. Some experienced worsening symptoms when using traditional medicine, though others used a combination of traditional and western medicine, employing traditional remedies to "*cool the sickness*" before going to hospital.

"*Interviewer1*: *Is that. . .is that usually what happens. . .that people maybe will try something. . .some traditional medicine first and then they will go to the hospital after*

*Respondent*: *Yeah many people do it to calm the tension*

*Respondent*: *To calm the tension of the sickness*

*Respondent2*: *To cool the sickness first*

*Interviewer1*: *To cool the sickness first*"

[FGD 1, four male participants]

## Discussion

Taken together, these qualitative and quantitative data show that sexual relationships between tourists and Gambian men working in the tourist industry are common, the transfer of money and resultant power dynamics in sexual relationships prevalent, and that men have psychological and sexual health and substance use care needs which are not currently being met. There is variation in risk and vulnerability within the sample, with men who have formal occupations such as tour guides or taxi drivers being less economically precarious, and less likely to report some risk behaviours. DCE data show that although the potential for partnerships to lead to travel to Europe is attractive to men, preferences are strong for shorter term transfers of money and consistent condom use with partners. Self-reported condom use was significantly more likely in sex acts with tourists than non-tourist partners.

These findings speak to a number of literatures. Evidence that men are willing to accept money for unprotected sex aligns with the broad literature on the economics of sex work [13–15, 25], though to our knowledge this is the first study to measure this phenomenon among heterosexual men providing transactional sex. That the health of Gambian men is influenced by the presence and behaviours of tourists aligns with literature on the political economy of tourism and globalization [26, 27], and empirical work demonstrating potential negative impacts of tourism of the health of residents [28], and on sexual health in particular [29, 30]. The findings of this study echo those of work with a population of men facing similar tourist-gender dynamics in Jamaica, highlighting the "numerous ways in which health intersects with issues of masculinity, sexuality and marginality" [10]. In addition, this work aligns with research among Mozambiquan women exploring sexual relationships with older European men and their influence on female migration which finds that "Sexual-monetary transactions, love, and desire must be understood as part of broader moralities of exchange in which migration to Europe and sending of remittances is also a kinship project" [31]. This study is unique in its sampling of a broad range of men who interact with tourists, which is a wider sample definition than the previous anthropologic study among this group [6]. This broad target population was intentional as preliminary work indicated that almost all men working with tourists may be looking for tourist friendships, and we therefore capture a fuller range of potential interactions by men at greater and lesser vulnerability. However, this wide target population may underestimate the prevalence of risk behaviours among the most risky or vulnerable, who–as elsewhere–may also be the hardest to recruit in studies which do not specifically target respondents on risk characteristics. Therefore, although these results may be generalisable to men working with tourists, they may understate the needs of those at highest risk or with greatest need.

All data were self-reported to interviewers in face-to-face interviews, and sensitive responses may be subject to acceptability biases. Although this may influence absolute estimates, within-study comparisons–for example comparing condom use with local and tourist partners–may be more valid if potential biases cancel out. We did not interview tourist partners with whom men formed relationships; alongside giving the perspectives of the tourist, data from partners could have validated measures of sexual risk behaviours. In addition, DCE data are obtained from respondents making choices between hypothetical alternatives. Although there is evidence that DCE choices correlate reasonably well with real-world choices [32], hypothetical bias remains a risk [33].

The context has changed since these data were collected in early 2019. Before the main tourist season at the end of 2019, Thomas Cook–the main package holiday firm bringing tourists to The Gambia–ceased operations, then in early 2020, Gambian borders were closed due to the SARS-CoV-2 pandemic. The impact of these significant and simultaneous shocks on men working with tourists, or the tourist industry more broadly, is not yet apparent.

Nevertheless, these results have a number of implications for the planning of preventative and curative healthcare services to meet the needs of *bumsters* and other men who have sexual relationships with tourists. First, money matters. Men with formal, registered jobs as taxi drivers and tour guides were perceived to have greater economic security and reported fewer risk behaviours than men in unregistered professions. In addition, household hunger–a key signal of substantial household poverty–was found correlated with drug use and STI symptoms. Furthermore, money was found to be a significant determinant of tourist partner preferences. It is critical, therefore, to enhance the economic security of this group of men, to reduce reliance on money from tourist relationships with power imbalances, with the intention to reduce risk behaviours.

Second, although physical health needs were largely met, we observe substantial unmet need for mental and sexual healthcare, and support for drugs, alcohol, and smoking. A first step for resolving this could be a mapping of services available to, and known by, men alongside further work to identify key barriers to care seeking. We did not collect data on reasons why care was not sought for health issues identified, but this information would be important to understand whether knowledge, access, or other issues result in men not seeking care. We did not explore in-depth the different contextual factors underlying the observed results or theorised how they might explain the reported results, but hypothesise that religion, colonial and post-colonial history and relationships, changing societal dynamics, technology, economic development and responsibilities of men, and gender-specific expectations in sexual and romantic relationships, will all be important factors. We recommend future researchers on bumsters in The Gambia explore these contextual factors.

## Conclusion

This study shows that sexual relationship between tourists and Gambian men are common, that these relationships offer both opportunity and risk to men, and that there are substantial mental and sexual health care and substance use needs in this group. The health system needs to understand barriers to existing health services, and how they can meet the needs of these vulnerable men.

## Supporting information

**S1 Text. Quantitative question scales.**
(DOCX)

**S2 Text. Qualitative topic guide for IDI and FGDs.**
(DOCX)

**S3 Text. Inclusivity in global research questionnaire.**
(DOCX)

**S1 Table. Additional results.**
(DOCX)

## Acknowledgments

We acknowledge the excellent research assistance of Mamudu Suso, Seedy Singateng, Ken Joof, and Kanagie Mankamang.

## Author Contributions

**Conceptualization:** Matthew Quaife, Melisa Martinez-Alvarez.

**Data curation:** Matthew Quaife, Mareme Diallo, Melisa Martinez-Alvarez.

**Formal analysis:** Matthew Quaife, Mareme Diallo, Melisa Martinez-Alvarez.

**Funding acquisition:** Matthew Quaife.

**Investigation:** Mareme Diallo.

**Supervision:** Assan Jaye, Melisa Martinez-Alvarez.

**Writing – original draft:** Matthew Quaife.

**Writing – review & editing:** Matthew Quaife, Mareme Diallo, Assan Jaye, Melisa Martinez-Alvarez.

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
