## [Decision Letter · Decision Letter 0]

16 Mar 2022

PGPH-D-21-00962

Partnership preferences, economic drivers, and health consequences of Gambian men's interactions with foreign tourists: A mixed methods study

Dear Dr. Quaife,

Thank you for submitting your manuscript to PLOS Global Public Health. After careful consideration, we feel that it has merit but does not fully meet PLOS Global Public Health’s publication criteria as it currently stands. Therefore, we invite you to submit a revised version of the manuscript that addresses the points raised during the review process.

We look forward to receiving your revised manuscript.

Kind regards,

Meghnath Dhimal, Ph.D.

Academic Editor

Journal Requirements:

1. Please include a complete copy of PLOS’ questionnaire on inclusiveness in global research in your revised manuscript. Our policy for research in this area aims to improve transparency in the reporting of research performed outside of researchers’ own country or community. The policy applies to researchers who have traveled to a different country to conduct research, research with Indigenous populations or their lands, and research on cultural artifacts. The questionnaire can also be requested at the journal’s discretion for any other submissions, even if these conditions are not met.  Please find more information on the policy and a link to download a blank copy of the questionnaire here: https://journals.plos.org/globalpublichealth/s/best-practices-in-research-reporting. Please upload a completed version of your questionnaire as Supporting Information when you resubmit your manuscript.

2. Please update your Competing Interests statement. If you have no competing interests to declare, please state: “The authors have declared that no competing interests exist.”

3. In the online submission form, you indicated that your data will be submitted to a repository upon acceptance. We strongly recommend all authors deposit their data before acceptance, as the process can be lengthy and hold up publication timelines. Please note that, though access restrictions are acceptable now, your entire data will need to be made freely accessible if your manuscript is accepted for publication. This policy applies to all data except where public deposition would breach compliance with the protocol approved by your research ethics board. If you are unable to adhere to our open data policy, please kindly revise your statement to explain your reasoning and we will seek the editor's input on an exemption. Please be assured that, once you have provided your new statement, the assessment of your exemption will not hold up the peer review process.

4. Please include the contents of your PGPH_abstract.docx file in your main text and remove it from your submission. The title page should not be uploaded as a separate file.

5. Please provide separate figure files in .tif or .eps format only and remove any figures embedded in your manuscript file. Please ensure that all files are under our size limit of 20MB.

6. We notice that your supplementary table is included in the manuscript file. Please remove them and upload them  with the file type 'Supporting Information'. Please ensure that all Supporting Information files are included correctly and that each one has a legend listed in the manuscript after the references list. 

7. Please provide us with a direct link to the base layer of the map used in Figure 1 and ensure this location is also included in the figure legend. 

Please note that, because all PLOS articles are published under a CC BY license (creativecommons.org/licenses/by/4.0/), we cannot publish proprietary maps such as Google Maps, Mapquest or other copyrighted maps. If your map was obtained from a copyrighted source please amend the figure so that the base map used is from an openly available source.

Please note that only the following CC BY licences are compatible with PLOS licence: CC BY 4.0, CC BY 2.0  and CC BY 3.0, meanwhile such licences as CC BY-ND 3.0 and others are not compatible due to additional restrictions. If you are unsure whether you can use a map or not, please do reach out and we will be able to help you. 

The following websites are good examples of where you can source open access or public domain maps:

Additional Editor Comments (if provided):

Reviewers' comments:

Reviewer's Responses to Questions

**Comments to the Author**

1. Does this manuscript meet PLOS Global Public Health’s publication criteria? Is the manuscript technically sound, and do the data support the conclusions? The manuscript must describe methodologically and ethically rigorous research with conclusions that are appropriately drawn based on the data presented.

Reviewer #1: Yes

Reviewer #2: Yes

2. Has the statistical analysis been performed appropriately and rigorously?

Reviewer #1: Yes

Reviewer #2: Yes

3. Have the authors made all data underlying the findings in their manuscript fully available (please refer to the Data Availability Statement at the start of the manuscript PDF file)?

Reviewer #1: Yes

Reviewer #2: Yes

4. Is the manuscript presented in an intelligible fashion and written in standard English?

Reviewer #1: Yes

Reviewer #2: Yes

5. Review Comments to the Author

Reviewer #1: This is an interesting study and well-written paper on the context of transactional sex between Gambian men and European female tourists. Using a simultaneous mixed method research design, the authors find that respondents working on the beaches of The Gambia face substantial health risks, including from STIs and mental health issues. The manuscript would be of interest to readers, and I have some major and minor questions/queries as well as suggestions to help the authors improve the clarity of their paper.

1. While I understand that there is a need to describe economic drivers and health implications of transactional interactions between Gambian men and female tourists, what are underlying theories that drive this study’s conceptual and methodological approach? As it stands, there is a lack of clarity on the theoretical grounding driving this important research. It seems that there is a need to develop a contextual understanding of certain practices have become/are integrated into the daily lives and interactions of Gambian men involved in the tourist industry. Qualitative results seem to show that in the participants’ descriptions of the economic drivers and the social contexts in which interactional sex along with alcohol and drug consumption occur, these are not seen as behaviors, but as deeply embedded social practices that are linked to broader issues of gender, sexuality, and economy.

2. It would also be beneficial to clarify the research design. The authors do a good job of presenting the methods used, but it would be helpful to understand how the methods and study design match up with the conceptual approach. Why a mixed-methods study design for the research questions? It is difficult to report on mixed methods and the way that the results are presented is a bit confusing. For instance, the study sample characteristics (quantitative survey) do not necessarily correspond with qualitative findings on perceptions of bumsters, economic drivers of this type of work, and relations with tourists. It might be helpful to organize the data in a way that makes more intuitive sense. One idea is to organize the results thematically from the qualitative results and this would require more in-depth discussion about study design.

3. It might be helpful to read others who have done work in the area of transactional sex between men and foreign female tourists, especially at the intersection of social science and global health. Some references include:

a. The work of Lauren Johnson: Johnson, L. C. (2016). ‘Men at risk’: sex work, tourism, and STI/HIV risk in Jamaica. Culture, health & sexuality, 18(9), 1025-1038.

b. Mark Padilla’s large body of work in the Caribbean: Padilla, M. (2008). Caribbean pleasure industry. University of Chicago Press; Colón-Burgos, J. F., & Padilla, M. (2021). Male transactional sex in the Dominican Republic: The politics of labor exclusion. In The Routledge Handbook of Male Sex Work, Culture, and Society (pp. 384-394). Routledge.

c. Emily Wentzell: Wentzell, E. (2014). ‘I help her, she helps me:’Mexican men performing masculinity through transactional sex. Sexualities, 17(7), 856-871.

d. Groes‐Green, C. (2014). Journeys of patronage: moral economies of transactional sex, kinship, and female migration from M ozambique to E urope. Journal of the Royal Anthropological Institute, 20(2), 237-255.

Reviewer #2: Review of the manuscript titled “Partnership, preferences, economic drivers and health consequences of Gambian men’s interactions with foreign tourists: A mixed methods study”.

This is an interesting study and a well-written paper. There are however some concerns that need to be clarified and revised for this paper to be published.

Abstract

In the abstract on page 1, the authors will need to include the actual sample size for the survey which was 244. They have noted that in the discrete choice experiment they had 208 respondents and this may erroneously be taken as the entire survey sample.

The authors write that about 49% of the Gambian men in their sample had sex with a tourist; however, in the descriptive table it shows 50%. This discrepancy should be rectified.

No results were stated on healthcare seeking behaviour among Gambian men in the abstract. However, there were recommendations about barriers to healthcare and health systems needs. Due to this, and the fact that assessing healthcare and health system needs are important to the study, the results on this topic should be shown in the abstract.

Introduction

On page 2, the authors suggest a number of studies have focused on the Western tourists’ sexual health needs and not on the host destination partners. These studies have to be cited and perhaps include a sentence summarizing what they state.

On page 3, the last-but-one paragraph under the Introduction summarizes the results but I think this needs to be removed from the Introduction.

Methods

Study setting

‘Tourist’ should be ‘tourists’ – page 3, paragraph 2, line 3.

Page 3, paragraph 2, line 7, the sentence should read ‘Although we anticipated that this approach was more sensitive than specific in identifying’. The word that should be ‘than’ and not ‘that’.

The map indicating the study setting and data collection locations needs to have a clearer way of pointing out the specific locations for the study as the circles are a bit big. Perhaps small stars, as was done in the inserted map of the Gambia would help.

Study design

The authors mentioned that they had a mixed method design with one qualitative component, but they rather had two qualitative components, since they carried out focus group discussions and in-depth interviews. This should be changed.

Information about the tools is in the study design sub-section which may not be needed since there is a section describing the tools. More description of the study design should be included. The authors state that the quantitative and qualitative data were collected simultaneously but analysed sequentially. The various steps used need to be explained in further detail.

Tool development

First use of LSHTM should be written out in full.

Sampling and data collection

Any information about pre-testing of the instruments should be stated.

More clarity needs to be made on the DCE framework when it comes to data collection. At the moment there is one section on quantitative survey implementation and nothing on the DCE so this needs to be stated. Were selected respondents from the survey taken through the DCE? What were the specific steps?

It should be mentioned in the discussion on sampling that it was purposive sampling that was undertaken.

Under the qualitative data collection heading there needs to be more information about the data collection. How were the 17 qualitative in-depth interview participants selected? Were all 17 in-depth interview participants chosen from the survey participants list, and was this at the interviewers’ discretion? Thus, did the respondents complete both qual and quant interviews? When were they selected? When were the focus group discussion participants selected? We also need information about the duration of these interviews and focus group discussions.

Data analysis

For the quantitative survey, the section describing the various independent variables needs to be moved to a separate section that explains how the variables were measured and why these were selected.

For the DCE, where there is a description about how the equations were estimated, there should be a brief statement about how the results were written up.

For the qualitative analyses, the authors begin by stating they developed a coding frame based on patterns in the quant data, suggesting these were analysed first. This suggests they carried out deductive coding and this must be stated more clearly. At the moment, they have only indicated they conducted inductive coding. Other information needs to be stated about the analyses, such as information about transcription (whether verbatim or not), whether the transcripts were checked, who in the research team carried out the analysis, if there was any reliability checks in terms of coding, how biases were reduced etc. The authors may use the standard reporting of qualitative research (SRQR) guidelines to help ensure they have documented all the information about the qualitative component of the study.

Results

For the descriptions of the bumsters section of the analyses, a description of the qualitative respondents must be stated somehow, whether through a table or summarized in the text. If no background characteristics were taken on the qualitative interview participants, reasons for this must be stated.

There are some quotes that mention the Gambian men engaging in relationships with male tourists. However, there is no discussion on this in the Introduction. There is a later mention of this in the manuscript, however, there could be a note on this somehow in the Introduction, so that readers understand that the Gambian men being studied may also have male partners. At the moment, we only get a sense of the tourists they engage in relationships with as only being women.

The quotes seem to have a few grammatical errors. There are also some quotes that may have been transcribed verbatim with the English not being understandable to all readers, for such quotes, the interpretations would have to be clearly stated in square brackets. This has been done to an extent, but the authors will need to have to go through all the quotes and ensure the essence of the statement can be understood.

The first quote on page 12 says “if you build and acquaintance” instead of “if you build an acquaintance”.

On page 14, first paragraph, line 3 – “A positive and significant” instead of “A positive ands significant”.

On page 16, the sentence under the drug and alcohol use paragraph, the third sentence says “exposes increase the risk of bumsters” . This should be revised.

For the healthcare needs and care-seeking sub-section of the Results, the figure should be moved to after the heading, and perhaps after some text.

On page 18, third paragraph, line 2, “some will prefer to die instead of talking about it” instead of “some will prefer to die inside of talking about it”.

The authors should clarify and make an argument as to why they reported on healthcare needs of all respondents and not just those that had sex with tourists. The interview guide also indicates that healthcare seeking behaviour questions were asked in general and not in relation to those that have sexual relationships with tourists. The rationale for this must be explained. If the healthcare needs in general are the desired result, then male healthcare seeking behaviours should not be interpreted as for only those who have sexual relationships with tourists but to men in general.

The authors refer to the sampling frame in the Methods and Results sections, however, it seems they are referring to the target population instead of the sampling frame because the sampling frame connotes that they were able to get a list of all members with the characteristics of interest and from this frame they sampled the respondents but this is not the sense we get from how their selection of respondents was carried out.

Discussion

The study seeks to indicate that the men are vulnerable due to their poverty, but is there any agency that these men have that can be explored and recommended?

Are there any existing policies in the Gambian tourism sector on protecting Gambians that can be mentioned and reinforced?

On page 19, the second paragraph, on the fourth line, the authors write “the health Gambian men” instead of “the health of Gambian men”.

On page 20 the first line it says “The study is unique its sampling” instead of “The study is unique in its sampling”.

On page 20, in the fifth paragraph, the last line, it says “or other issues result in men” instead of “or other issues resulting in men”.

Conclusion

The study does not necessarily provide a sense of the opportunities tourists gain from sexual relationships with their Gambian partners.

Other general comments

The Vancouver reference style was not used as prescribed in the authors’ guidelines for submissions. The authors also included the abstract on the title page which is not necessary.

6. PLOS authors have the option to publish the peer review history of their article (what does this mean?). If published, this will include your full peer review and any attached files.

**Do you want your identity to be public for this peer review?** For information about this choice, including consent withdrawal, please see our Privacy Policy.

Reviewer #1: No

Reviewer #2: No

---

## [Decision Letter · Decision Letter 1]

27 Dec 2022

Partnership preferences, economic drivers, and health consequences of Gambian men's interactions with foreign tourists: A mixed methods study

PGPH-D-21-00962R1

Dear Dr. Quaife,

We are pleased to inform you that your manuscript 'Partnership preferences, economic drivers, and health consequences of Gambian men's interactions with foreign tourists: A mixed methods study' has been provisionally accepted for publication in PLOS Global Public Health.

Best regards,

Meghnath Dhimal, Ph.D.

Academic Editor

Reviewer Comments (if any, and for reference):

Reviewer's Responses to Questions

**Comments to the Author**

1. If the authors have adequately addressed your comments raised in a previous round of review and you feel that this manuscript is now acceptable for publication, you may indicate that here to bypass the “Comments to the Author” section, enter your conflict of interest statement in the “Confidential to Editor” section, and submit your "Accept" recommendation.

Reviewer #2: All comments have been addressed

2. Does this manuscript meet PLOS Global Public Health’s publication criteria? Is the manuscript technically sound, and do the data support the conclusions? The manuscript must describe methodologically and ethically rigorous research with conclusions that are appropriately drawn based on the data presented.

Reviewer #2: Yes

3. Has the statistical analysis been performed appropriately and rigorously?

Reviewer #2: Yes

4. Have the authors made all data underlying the findings in their manuscript fully available (please refer to the Data Availability Statement at the start of the manuscript PDF file)?

Reviewer #2: No

5. Is the manuscript presented in an intelligible fashion and written in standard English?

Reviewer #2: Yes

6. Review Comments to the Author

Reviewer #2: The authors have done a great job of revising the manuscript. I have no further comments except one suggetion that the authors check the main sub-headings under the Results so that they truly reflect the four main topics they stated would be discussed. The additional sub-headings can be included underneath these, where needed.

7. PLOS authors have the option to publish the peer review history of their article (what does this mean?). If published, this will include your full peer review and any attached files.

**Do you want your identity to be public for this peer review?** For information about this choice, including consent withdrawal, please see our Privacy Policy.

Reviewer #2: No
